# Diversification of *Pseudomonas aeruginosa* Biofilm Populations under Repeated Phage Exposures Decreases the Efficacy of the Treatment

**DOI:** 10.3390/microorganisms12091880

**Published:** 2024-09-12

**Authors:** Mark Grevsen Martinet, Mara Lohde, Doaa Higazy, Christian Brandt, Mathias W. Pletz, Mathias Middelboe, Oliwia Makarewicz, Oana Ciofu

**Affiliations:** 1Institute of Infectious Diseases and Infection Control, Jena University Hospital, 07747 Jena, Germany; mark.martinet@med.uni-jena.de (M.G.M.); mara.lohde@med.uni-jena.de (M.L.); christian.brandt@med.uni-jena.de (C.B.); mathias.pletz@med.uni-jena.de (M.W.P.); oliwia.makarewicz@med.uni-jena.de (O.M.); 2Costerton Biofilm Center, Department of Immunology and Microbiology, Faculty of Health and Medical Sciences, University of Copenhagen, 2200 Copenhagen, Denmark; doaa.mohamed@agr.cu.edu.eg; 3Leibniz Center for Photonics in Infection Research (LPI), 07745 Jena, Germany; 4Marine Biological Section, Department of Biology, University of Copenhagen, 3000 Helsingør, Denmark; mmiddelboe@bio.ku.dk; 5Department of Biology, University of Southern Denmark, 5230 Odense, Denmark

**Keywords:** bacteriophages, biofilms, *Pseudomonas aeruginosa*, diversification of population, genomics, motility

## Abstract

Phage therapy has been proposed as a therapeutic alternative to antibiotics for the treatment of chronic, biofilm-related *P. aeruginosa* infections. To gain a deeper insight into the complex biofilm–phage interactions, we investigated in the present study the effect of three successive exposures to lytic phages of biofilms formed by the reference strains PAO1 and PA14 as well as of two sequential clinical *P. aeruginosa* isolates from the sputum of a patient with cystic fibrosis (CF). The Calgary device was employed as a biofilm model and the efficacy of phage treatment was evaluated by measurements of the biomass stained with crystal violet (CV) and of the cell density of the biofilm bacterial population (CFU/mL) after each of the three phage exposures. The genetic alterations of *P. aeruginosa* isolates from biofilms exposed to phages were investigated by whole-genome sequencing. We show here that the anti-biofilm efficacy of the phage treatment decreased rapidly with repeated applications of lytic phages on *P. aeruginosa* strains with different genetic backgrounds. Although we observed the maintenance of a small subpopulation of sensitive cells after repeated phage treatments, a fast recruitment of mechanisms involved in the persistence of biofilms to the phage attack occurred, mainly by mutations causing alterations of the phage receptors. However, mutations causing phage-tolerant phenotypes such as alginate-hyperproducing mutants were also observed. In conclusion, a decreased anti-biofilm effect occurred after repeated exposure to lytic phages of *P. aeruginosa* biofilms due to the recruitment of different resistance and tolerance mechanisms.

## 1. Introduction

Bacterial populations embedded in the extracellular matrix of biofilms are characterized by a diversity of metabolic states determined by the life in gradients of nutrients and oxygen and have a high degree of heterogeneity. The organization in these multicellular structures ensures tolerance to the immune system and antimicrobial agents of the biofilm bacterial populations. In addition, the development of genetic resistance is also promoted in biofilms [1].

Causing a large variety of chronic, persistent infections, biofilms represent a therapeutic challenge. Despite intensive research in anti-biofilm strategies, a limited number have reached clinical trials, and the actual treatment of biofilm infections is still based on a combination of antibiotics, in high doses and for long periods [2]. Management of these difficult-to-treat infections is further complicated by the occurrence of multi-drug-resistant pathogens in some clinical settings and requires new therapeutic alternatives. In this context, bacteriophages and viruses that infect bacteria have been considered as one of the attractive solutions [3]. Recent case reports of phage therapy as a last resort treatment, when antibiotics were insufficient, have shown promising effects of phage therapy on *P. aeruginosa* lung colonization [4]. In many cases, these infections were caused by biofilms, and the phages have been administered in repeated dosages. Although applied compassionately in the clinical praxis, the biology of the phage–biofilm interactions is very complex [5,6,7] and not very well understood. Many of the mechanisms responsible for the tolerance of biofilms to antibiotics also play a role in the tolerance of biofilms towards lytic phages [8].

We have previously reported in flow-cell biofilms of PAO1 that repeated treatments with lytic phages selected for larger biofilms were less effective in biofilm eradication than single treatments, suggesting that biofilm size and complexity are positively correlated with phage infection pressure [9].

The promotion of biofilm formation under phage treatment has been shown also by other studies [10]. However, due to the high specificity of bacteria–phage interactions, making general predictions about the effect of repeated phage therapy on biofilms is difficult.

To gain a deeper insight into biofilm–phage interactions, we decided to investigate the effect of repeated phage exposure to biofilms using various *P. aeruginosa* strains and lytic phages. The efficacy of phage treatment was evaluated by the reduction in the biomass as measured by crystal violet (CV) staining and of the cell density of the biofilm bacterial population (CFU/mL) after each of the three phage exposures. As these parameters depend on the tolerance and the resistance of biofilms to phage attack, we were also interested in phenotypical and genetic characterization of isolates from the biofilm populations that evolved under phage treatment. The phenotypic diversity of the bacterial populations was characterized in terms of degree of sensitivity to the lytic phages, colony morphology, growth curves, and twitching motility.

We show here that the anti-biofilm efficacy of the phage treatment decreased with repeated applications of NP3 phage [11] on four *P. aeruginosa* strains with different genetic backgrounds: two *P. aeruginosa* reference strains PAO1 and PA14 [12] and two sequential clinical *P. aeruginosa* isolates from the sputum of a patient with cystic fibrosis (CF). Similar effects were observed with repeated treatments of the two *P. aeruginosa* reference strains with other lytic phages such as PA14 biofilms with NP1 and PAO1 biofilms with vB_Pae-TbilisiM32, suggesting a common evolutionary trend.

It has been predicted that spatially structured communities promote the protection of sensitive host cells from phage exposure and thus decrease the emergence of phage resistance [13]. Although we observed the maintenance of a small subpopulation of sensitive cells after repeated phage treatments, a fast recruitment of mechanisms involved in the persistence of biofilms to the phage attack occurred represented by a selection of resistant and tolerant mutants. The resistance mechanisms were represented mainly by mutations causing alterations of the phage receptors or impairing the phage infection, but mutations causing phage-tolerant phenotypes such as alginate-hyperproducing mutants were also observed.

## 2. Material and Methods

### 2.1. Bacterial Strains and Phages

*P. aeruginosa* reference strains PAO1gfp (Tn7-strep) [9] and PA14gfp [14] were used. These two common laboratory strains of *P. aeruginosa* represent two distinct lineages based on whole-genome phylogenetic analysis [15]. The green fluorescence protein (GFP) labeling of the strains was only used for control reasons to verify that the phage-resistant clones did not contain contaminants.

Two *P. aeruginosa* isolates from one cystic fibrosis (CF) patient, who was intermittently colonized with isolate CF341_06 in 2006 and chronically infected with isolate CF341_08 in 2008 [16] were used to explore whether the phages exhibit similar effects on clinical isolates. The clinical isolates were collected from the sputum of the patient and stored at −80 °C in serum broth with 10% glycerol as a cryoprotectant (SSI Diagnostica, Hillerød, Denmark) and are part of the bacterial collection available at the Department of Clinical Microbiology, Rigshospitalet. Additional information on the two CF clinical isolates is presented in Appendix A.

Phage NP3 belonging to Pbunaviruses (GenBank accession number KU198331) [11] was made available from Prof. Bruce Levin’s laboratory, Emory University (Atlanta, GA, USA). This phage was isolated from sewage in Atlanta, Georgia, USA. It has been shown that NP3 has a lytic effect against both PAO1 and PA14 and a broad host range, as evaluated on several *P. aeruginosa* isolates from the environment and CF patients using the spot test on double-layer agar (Appendix A).

Phage NP1 belonging to Nipunaviruses was provided from the same laboratory as NP3 (GenBank accession number KX129925) with lytic activity against PA14 but not against PAO1, and phage vB_Pae-TbilisiM32 (thereafter named M32), provided by the Phage Therapy Center in Tbilisi (Georgia) (GenBank accession number KX711710.1), with lytic activity against PAO1 but not PA14, was also used. Based on our own sequencing data, phage M32 is from the phiKMV-like subgroup of Podoviruses and was previously suggested to recognize type IV pili as a possible receptor on the bacterial surface [17]. According to Chaudhry et al. [11], the host receptors for NP3 are suggested to be the lipopolysaccharide core and the common antigen.

All phages were sequenced and analyzed (see below) and showed high DNA sequence identities to the corresponding reference genomes: NP1 82.68%, NP3 93.29%, and M32 99.85%.

### 2.2. Sequencing of Phages

All three phages have been sequenced before being used in this study, to ensure their identity. Briefly, 1 mL of phage aliquot was firstly treated with 10 U of DNase I (Thermo Fisher Scientific, Waltham, MA, USA) for 15 min at room temperature, followed by DNase I inactivation at 75 °C for 5 min in a thermoblock (Thermo Fisher Scientific). Protein residues were digested by the addition of approximately 6 U of Proteinase K (Thermo Fisher Scientific) per aliquot and incubated at 55 °C for 30 min. The DNA was then purified using ZymoBiomics Genomic DNA Clean & Concentrator (both ZymoResearch, Freiburg, Germany) according to the manufacturer’s protocol. The concentration was assessed using Qubit™ 1X dsDNA HS Assay Kit (Thermo Fisher Scientific) according to the manufacturer’s protocol.

DNA was pre-washed and size-selected by using 0.45× volume of Ampure XP (Beckman Coulter, Brea, CA, USA) and eluted in 50 μL DNase-free water. Libraries were prepared from 1000 ng input DNA using the SQL-LSK110 kit (Oxford Nanopore Technologies, Oxford, UK) according to the manufacturer’s protocol. Incubation times for end repair, dA-tailing, and ligation were increased (doubled) to improve ligation efficiency. The DNA libraries were sequenced using FLO-MIN106 R9.4.1 flow cells on GridION using MinKNOW software 5.5.10 (all Oxford Nanopore Technologies), the standard 72 h runscript, and the high-accuracy basecalling model and demultiplexing mode.

### 2.3. Propagation and Titration of Phages

To produce fresh phage lysates for the experiments, the phages NP3 and NP1 were propagated in PA14, and vB_Pae-TbilisiM32 was propagated in PAO1. The phages were serially diluted, and 100 µL was mixed with the respective host bacteria (OD_600_ of 0.1) and incubated for 30 min at 37 °C without shaking. Thereafter, 4 mL LB agar (0.5%) was added, and the suspension was gently mixed and immediately overlayed on LB agar (2%) and incubated overnight at 37 °C. To isolate the phages, 5 mL LB medium was added to plates with confluent lysis and incubated for 2 h at room temperature and shaking (70–130 rpm). The LB medium was then filtered through a 0.22 μm filter (TPP, Trasadingen, Switzerland), and the phage stock was stored at 4 °C. Every new lysate was titrated using the same procedure and the plaques were counted on plates where 30 to 300 plaques were visible and expressed as plaque-forming units per milliliter (PFU/mL).

### 2.4. Biofilm Formation

The static biofilm model formed in the modified Calgary device was employed in this short experimental evolution study [18]. The strains were grown in LB broth overnight. The overnight cultures of each strain were diluted to OD_600nm_ of 0.01 in LB broth, and 120 µL was added to each well of the Nunc™ MicroWell™ 96-Well Microplates (Thermo Fisher Scientific). The plates were covered by lids containing 96 pegs (Nunc-Transferable Solid Phase screening system (Nunc-TSP, Thermo Fisher Scientific)), which submerged into the bacterial cultures in the wells and were incubated overnight at 37 °C to form biofilms on the Nunc-TSPs of the lid.

### 2.5. Phage Treatment

The Nunc-TSP lid containing the biofilm was rinsed three times in saline and transferred to a new Nunc™ MicroWell™ 96-Well Microplate containing the phages at different concentrations (depending on experiment) in LB media.

To investigate the effects of the phage concentration on biofilms, the phages NP1 and NP3 were serially diluted, and 120 µL of phage suspensions with concentrations from 10^1^ PFU/mL to 10^10^ PFU/mL were added per well in half of the plate (48 wells), and LB media (controls) were added in the other half (48 Nunc-TSPs). For repeated phage treatment, 10^8^ PFU/mL of the phages were used per well, and the Nunc-TSPs covered with biofilm were transferred to new plates containing fresh phage suspension after 24 and 48 h phage treatment, resulting in 24 h (single), 48 (two), and 72 (three) treatments. The biomass and the size of the biofilm bacterial population (CFU/mL) were assessed in the treated and control biofilms after each transfer. Half of the biofilms (24 Nunc-TSPs) were used for biomass determination and the other half (24 Nunc-TSPs) for CFU determinations.

All experiments were performed at least in two biological and two technical replicates.

### 2.6. Crystal Violet Staining

To determine the biomass, the biofilm Nunc-TSPs were placed into a new 96 Nunc™ MicroWell™ 96-Well Microplate containing 120 µL of a 0.1% crystal violet (CV) solution, and the biofilms were stained for 15 min. The stained biofilm Nunc-TSPs were washed 3 times in fresh 96-well plates containing water with a concentration of NaCl of 0.9% each and finally placed into a fresh 96-well plate containing 32% acetic acid (de-staining) for 15 min. The released CV into the acetic acid was quantified by absorbance measurements at 590 nm in the high-performance filter-based multimode microplate reader Victor Nivo (Perkin Elmer, Waltham, MA, USA). Background absorbance of empty wells (baseline) was subtracted from all values.

### 2.7. Viable Bacteria Count

To determine the size of the biofilm population, the biofilms were disrupted from the Nunc-TSPs by sonication of the pegs in 120 µL LB for 30 s in the Branson 1510 ultrasonic bath (Sigma-Aldrich, St. Louis, MO, USA). The bacterial suspension obtained after sonication was serially diluted, and 20 µL of dilutions between 10^−2^ and 10^−5^ were plated on 2% LB agar plates and incubated overnight at 37 °C. LB plates containing 30 to 300 colonies were counted, and the viable bacteria fraction was quantified as colony-forming units per milliliter (CFU/mL).

After one, two, and three rounds of phage exposures, 48 randomly selected colonies (16 from each treatment) were isolated from each species/phage pair from the CFU counting plates. Overnight cultures in LB of the isolated colonies were frozen at −80 °C in 96-well microtiter plates for further analysis. Before phenotypic analysis, the colonies were passed three times through phage-free LB media (plates and broth). The colony morphology was observed, and the overproduction of alginate (mucoid) or pigment production (pyomelanin) was documented.

### 2.8. Twitching Motility

The twitching motility was examined by inoculating single colonies in the bottom of 1% LB agar followed by incubation at 37 °C and measurement of the size (diameter) of the bacterial growth from the inoculation spot after 48 h. Visible growth on the spot without any spread was assessed as zero.

### 2.9. Growth Experiments

Overnight LB cultures of the clones after phage treatment were adjusted to OD_600_ of 0.1, diluted 1000-fold, and 100 µL of each suspension was transferred in triplicate into a 96-well Nunc plate with a flat bottom (Thermo Fisher Scientific). The growth was recorded as OD_600_ in an Infinite F200 Pro plate reader (Tecan, Männedorf, Switzerland) at 37 °C at 225 rpm for 24 h every 20 min using Magellan V 7.2 software (Tecan). Growth curves were constructed by subtracting the baseline (medium only); growth rate was assessed for log phase.

### 2.10. Phage Susceptibility Assay

The screening spot assay was performed to differentiate phage-sensitive and resistant clones from the biofilm populations after phage treatment by spotting 5 µL of undiluted respective phage stock (10^11^ PFU/mL) on LB agar plates covered with 4 mL soft agar (0.5% agar) containing 0.3 mL of the individual host bacteria from a mid-log phase (optical density OD_600_ of 0.1). The plates were incubated at 37 °C, and after 24 h, the sensitivity to the phage was determined based on the presence of the visible inhibition halo in the bacterial lawn. After these investigations, the colonies were divided into sensitive (formation of inhibition halo) and resistant (no inhibition) colonies. The group of sensitive colonies were further analyzed for their sensitivity to serial phage dilutions to determine the phage efficacy by PFU/mL.

### 2.11. Sequencing of the Bacteria

Stochastically selected colonies from repeated phage treatment experiments were whole-genome-sequenced by Illumina technology (Eurofins Genomics Europe Shared Services GmbH, Ebersberg, Germany). To optimize resource use given the project’s budget constraints, a strategic approach was employed in selecting colonies for molecular analysis. Sequencing was conducted on a maximum of 8 resistant and at least 3 sensitive colonies per group. Initial sequencing of multiple sensitive colonies from the M32 strain revealed consistent results, allowing us to limit further sequencing to three sensitive colonies without compromising data quality. Challenges were encountered in obtaining sufficient representatives from some groups, particularly the sensitive colonies in PAO1 and NP3, or in achieving high-quality sequence data, which influenced the final selection of colonies for analysis.

In total, 9 colonies of NP3-treated and 16 colonies of M32-treated PAO1 biofilms, 12 colonies of NP3-treated PA14 biofilms, as well as 9 colonies from each of the NP3-treated CF 341_06 and CF341_08 biofilms were sequenced. Additionally, untreated biofilms of PAO1 and PA14 were grown for 72 h with daily media change, and 10 colonies from each strain were sequenced as controls. The ancestor strains PAO1, PA14, CF 341_06, and CF 341_08 were also sequenced and used as reference genomes to identify mutations in the clones.

The DNA was isolated from overnight cultures of the respective strains using the ZymoBiomics DNA Microprep (Zymo Research, Freiburg, Germany) according to the manufacturer’s instructions. The concentration was assessed using Qubit™ 1X dsDNA HS Assay Kit (Thermo Fisher Scientific) according to the manufacturer’s protocol.

### 2.12. Illumina Sequencing, Assembly, and Annotation

The DNA of the ancestor strains and their clones derived after repeated treatment with phage NP3 or M32, and the respective untreated clones after 72 h growth were isolated using the ZymoBIOMICS DNA Mircoprep Kit (Zymo Research Europe GmbH, Freiburg im Breisgau, Germany) according to the manufacturer’s instructions. DNA concentration was determined by using the Invitrogen™ Qubit™ 1X dsDNA Assay Kit on the Qubit4 fluorometer (both Thermo-Fischer Scientific) according to the manufacturer’s instruction. DNA sequencing by Illumina technology was outsourced to Eurofins Genomics (Eurofins Genomics Germany GmbH, Ebersberg, Germany).

De novo assembly of the Illumina-sequenced reference strains CF06, CF08, PAO1, and PA14 was performed by SPAdes v3.15.5 [19] using the careful-option flag and annotated first using prokka v1.14.6 [https://github.com/tseemann/prokka, accessed on 1 November 2023]. Contigs below 1000 bases were excluded from further analysis. The altered genes that were classified as hypothetical by prokka were further annotated using bakta v1.7.0 [20].

### 2.13. Sequence Analysis and Variant Calling

Reads were basecalled using Guppy v4.2.2 GPU basecaller (Oxford Nanopore Technologies) during sequencing using the high-accuracy basecalling model.

The phage genomes were reconstructed using Flye v2.9.3, Racon v1.4.20, and Medaka v1.11.3. WtP v1.2.3 (https://mult1fractal.github.io/wtp-documentation/, accessed on 3 November 2023) was used to confirm phage sequences [21]. The NCBI nucleotide Blast search was used to compare the phage sequences with existing phages.

To examine mutations resulting from phage treatment or prolonged growth in the *P. aeruginosa* clones, pairwise variant calling was conducted using snippy v4.6.0 [https://github.com/tseemann/snippy, accessed on 3 November 2023] by comparing the FASTQ files of the isolates to the previously assembled and annotated respective ancestor strains (GBK files).

### 2.14. Statistical Analysis

The effects of phages on cell viability and absorbance were visualized and analyzed using GraphPad Prism 9.4.1. (GraphPad Software, San Diego, CA, USA). For statistical analysis, the nonparametric Kruskal–Wallis test and Dunn’s multiple comparisons post-test were used; significance was assumed for *p*-values < 0.05 (two-sided confidence intervals 5–95%).

The statistics for differences in phenotypic parameters between resistant and sensitive clones from each strain/phage pair experiment were analyzed by unpaired t-tests with unequal variance in Excel for Microsoft 365 (Microsoft Corporation, Redmond, WA, USA).

## 3. Results

### 3.1. The Anti-Biofilm Effect Is Phage-Concentration-Dependent

The effect of phage concentration on *P. aeruginosa* PAO1 and PA14 biofilm reduction was determined by the quantification of viable bacteria and crystal violet staining after exposure of 24 h old biofilms formed on Nunc-TSPs to serially diluted NP3 and NP1 phages for further 24 h and compared to the control biofilms grown under the same conditions without phages (Figure 1).

To assess the minimal biofilm eradication concentrations (MBEC), defined in this study as the phage concentration that caused a 99.9% eradication of the biofilm-embedded bacterial population (3 log_10_ reduction in CFU/mL compared to untreated biofilms), the viable bacteria (CFU/mL) of the biofilm population were transformed as a percentage of viable cell count relative to the untreated controls (Figure 1A). There was a clear correlation between phage concentration and biofilm killing, but poor reduction in the biofilm populations was observed at phage concentrations below 10^3^ PFU/mL. A stronger reduction was achieved when the biofilms were exposed to phage concentrations higher than 10^5^ PFU/mL for NP3 and NP1 in PA14 and 10^7^ PFU/mL for NP3 in PAO1. The MBEC_NP3_ and MBEC_NP1_ were both achieved at 10^8^ PFU/mL in PA14, while in PAO1 the MBEC_NP3_ was not achieved even at 10^10^ PFU/mL.

The crystal violet staining, which represents an indirect method by staining the negatively charged exopolymers of the biofilm matrix, indicated that the biofilm mass was reduced already at a phage concentration of 10^1^ PFU/mL and that a higher NP3 phage concentration did not result in a significantly stronger biomass reduction (Figure 1B).

### 3.2. The Anti-Biofilm Effect Decreases with Repeated Phage Treatments

Based on the previous results, we used 10^8^ PFU/mL for the repeated treatments of the biofilm of PAO1, PA14, and two CF isolates, CF341_06 and CF341 NM_08, with the different phages, even if this phage concentration was below the MBEC_NP3_ of PAO1. In both methods, the CFU/mL and crystal violet staining were applied to visualize the effects of repeated phage treatments with fresh phage lysates for 3 × 24 h (corresponding to 24 h, 48 h, and 72 h).

In the untreated biofilms (Figure 2, Figure 3 and Figure 4), the biofilm mass and CFU number increased visibly (and in most cases significantly) within the experimental time corresponding to additional 24 h, 48 h, and 72 h of biofilm growth. When exposing the biofilms to phages, a significant reduction in the biomass was observed, and the densities of viable bacteria were reduced by a least 2 logs to 10^2^–10^6^ CFU/mL for all *P. aeruginosa* strains after the first 24 h of phage treatment (Figure 2, Figure 3 and Figure 4). This initial decrease in response to phage treatment was followed by a stepwise increase in biofilm mass and viable bacteria count with repeated treatments independent of the phage. After 72 h (third treatment), the control biofilms and the treated biofilms had reached the same level in both mass and number of viable bacteria, and there were no significant effects of phage exposure (Figure 2, Figure 3 and Figure 4).

### 3.3. Sensitivity to Phages of Clones from Phage-Treated Biofilms

From all the experiments, 16 randomly selected colonies from each treatment (a total of 48 colonies) were collected and tested for phage susceptibility (Table 1). After treatment with NP3, all the isolated PAO1 clones were NP3-resistant, the percentage of resistant clones was 79.2% for PA14, 70.8% for CF341_06 and 66.7% for CF341_08. After treatment with M32, 83.3% of the PAO1 clones were resistant to M32 (Appendix A).

### 3.4. Twitching Motility of the Clones after Treatment with Phages

Type IV pili have been recognized as phage receptors [22] and were suggested as receptors for M32. Type IV pili are involved in twitching motility, and we therefore decided to investigate this type of motility in the collected isolates.

Comparing the twitching motility of the phage-treated clones to untreated clones, we observed that clones isolated after phage treatment had a reduced twitching motility compared to untreated clones. A significantly reduced twitching motility of resistant clones compared to the phage sensitive clones was observed for the PA14 clones obtained after treatment with NP3. No significant differences between sensitive and resistant clones were observed for PAO1 treated with M32 and for CF341_06 treated with NP3. The CF341_08 ancestor had a reduced twitching motility compared to the reference *P. aeruginosa* strains and the earlier CF 341-06 isolate, making it difficult to evaluate the changes in motility for the clones isolated from this strain (Appendix A).

### 3.5. Effect of Phage Treatment on Growth Rate

As mutations causing resistance might have a fitness cost, we decided to measure the growth rates of the collected clones.

Measurement of the growth rates in LB of the sequenced isolates showed no difference between the ancestor and the phage-exposed clones (Table 1). Lower growth rates were observed for some of the clones collected from biofilms of *P. aeruginosa* 341_06 (isolates S_H1 and R_H3) and 341_08 (all sequenced isolates except R_F8 and S_F7), suggesting a fitness cost of the mutations acquired in these specific clones during evolution in the presence of phages. A significantly slower growth rate was observed in NP3-resistant PA14 isolates compared to sensitive ones. (Appendix A).

### 3.6. Effect of Phage Treatment on Colony Morphotypes

Mucoid clones were already observed in PAO1 biofilms exposed to M32 (1 clone) and in PA14 exposed to NP3 (3 clones) after two rounds of exposures to phages. Pyomelanin production was observed in clones from biofilms exposed to NP3 of PA14 (seven clones), PAO1 (one clone), and CF 341_06 (two clones) and after exposure to M32 of PAO1 biofilms (1 clone) (Appendix A).

### 3.7. Spontaneous Genetic Alterations in Clones Isolated from Biofilms

To identify spontaneous mutations, ten randomly selected colonies of PAO1 and PA14 biofilms, cultivated on the modified Calgary device for 72 h without phage treatment, were subjected to sequencing and subsequently compared to the sequences of their respective ancestors. No alterations were discerned within the 72 h timeframe for PA14. In four clones of PAO1, however, an amino acid substitution, specifically Arg134Ser in the *siaB* gene encoding a kinase, was detected (Figure 5F). The observed substitution at the C-terminus of the α5 helix in SiaB does not occur within the purportedly active amino acids [23], but this does not exclude the fact that the protein function is not affected. The SiaA/B/C/D signaling network regulates intracellular c-di-GMP levels in *P. aeruginosa*, subsequently influencing cellular aggregation and biofilm formation. Considering that this alteration did not manifest under phage treatment within the analyzed clones, it can be inferred that it lacks any association with phage resistance and, consequently, should be classified as a spontaneous mutation occurring in biofilms.

### 3.8. Genetic Alterations Related to Exposure by M32 and NP3 Phages

The genetic changes found in the sequenced clones are shown in Figure 5 and in more detail in Appendix A.

From biofilms of PAO1 treated with NP3, a total of nine clones, three from each time-point, have been sequenced (Figure 5A). No genetic changes have been identified in two of the clones. From biofilms of PA14 treated with NP3, twelve clones have been sequenced, with four from biofilms treated once, four from biofilms treated twice, and four from biofilms treated thrice. No genetic changes have been identified in three of the clones (Figure 5B). From biofilms of PAO1 treated with M32, sixteen colonies have been sequenced, four from biofilms treated once, six from biofilms treated twice, and six from biofilms treated thrice. Genetic changes were identified in all clones (Figure 5C). From biofilms of CF341_06 and CF341_08 treated with NP3, nine clones have been sequenced, three from each time-point, and genetic alterations have been identified in all clones (Figure 5D,E).

In the phage-resistant clones, mutations were predominantly identified in individual genes associated with type IV pilus biosynthesis and function (green bars) and LPS biosynthesis (yellow bars).

Mutations in genes involved in type IV pilus biosynthesis, such as a frameshift at Ser173 in the gene encoding PilX, were identified in two M32-resistant PAO1 clones. Additionally, one NP3-resistant PAO1 clone carried a nonsense mutation at Gln148 in PilX, and another NP3-resistant clone had a mutation at Glu340 in PilS. In a PAO1 clone resistant to NP3, a substitution Thr117Pro occurred in PilE. PilX and PilE are two of the five minor fiber pilins (FimU-PilVWXE) [24]. PilS is the sensor kinase of the two-component system PilS-PilR, regulating the expression of the type IV pilus major subunit PilA. In an NP3-resistant PAO1 clone, a Val642Gly substitution was identified in a loop situated within the inner site of the structural protein PilQ. In two NP3-resistant PA14 clones, a frameshift mutation with a likely loss of function was observed in PilB, while one NP3-resistant PAO1 clone established a substitution, Asp388Ala, in PilB. PilB is a component of the intracellular motor sub-complex in the type IV pilus assembly machinery.

In one M32-resistant PAO1 clone, a substitution Arg120Cys occurred in the 3′,5′-cyclic-AMP (cAMP) phosphodiesterase (CpdA) with unknown consequence. The intracellular second messenger cAMP levels can be induced by the type IV pilins subunit PilA and PilT [25], and therefore, we related the CpdA mutation to the type IV pilus function.

In an NP3-sensitive clone of CF341_06, a Gly470Cys substitution was detected in PilQ. This exchange from a neutral backbone amino acid Gly to a functional and SH-reactive Cys residue within the membrane-located β-sheets might disrupt pore symmetry and lead to dysfunction. In one NP3-sensitive CF341_08 clone, an Arg84Trp substitution occurred in PilN, one of the inner membrane-anchored proteins of the type IV pilus assembly machinery [24].

In all NP3-resistant and M32-resistant clones, but also four M32-sensitive clones of PAO1, mutations were found in the genes *wzy*. A frameshift at Thr 46 was the most prominent alteration found in all but two phage-resistant clones and two sensitive clones. An Arg176Cys substitution was found in one M32-resistant and two M32-sensitive PAO1 clones, while a nonsense mutation in Arg212 was present in one M32-resistant PAO1 clone. In one M32-sensitive PAO1clone, a frameshift in the *wzzB* gene was identified. Both Wzy and WzzB are components of the same metabolic pathway and regulate the O-antigen chain length [26,27].

Different mutations in the *galU* gene, which encodes for UTP-glucose-1-phosphate uridylyltransferase GalU, were observed in two NP3-resistant CF341_06 clones and one PA14 clone (Figure 5B,D). GalU plays a role in lipopolysaccharide core region biosynthesis. In two clones, frameshifts due to deletions occurred at amino acid positions Leu6 or Met105 in GalU. In the other clone, a Gly170Ala substitution was present.

Regarding RfaB-like glycosyltransferase, which is involved in peptidoglycan and LPS synthesis [28], different mutations accumulated in NP3-sensitive and -resistant clones of PA14, CF341_06, and CF341_08. In the sensitive CF341_06 clone, the Phe154Ile substitution was identified, potentially weakly impacting protein function. In the sensitive PA14 clone, RfaB was truncated at amino acid position 310, likely resulting in loss of function. In an NP3-resistant CF341_08 clone, a frameshift at the last C-terminal amino acid (Cys378) in RfaB prolonged the C-terminus. In an NP3-resistant PA14 clone, a frameshift at amino acid position 55 occurred.

In four NP3-resistant and two NP3-sensitive clones of the CF341_08 strain, a deletion of 20 nucleotides occurred at amino acid position Lys235 in the *dnpA* gene that is a putative de-N-acetylase and belongs to the LPS biosynthesis cluster [29]. It seems that this mutation occurred early in the phage-exposed biofilm population and accompanied the alteration of the *dnaX* gene in five clones.

Frequent mutations in the *rluA* gene that encodes for the rRNA and tRNA-specific pseudouridine synthase RulA seemed not to be related to the phage resistance as they were found in almost all M32-sensitive clones of PAO1 (Figure 5C).

Various frameshifts and nonsense mutations were discovered in the *mucA* genes, resulting in a loss of function. These mutations were identified in M32- and NP3-sensitive clones of both PAO1 and PA14 and in only one NP3-resistant clone of PA14. MucA serves as an anti-sigma factor, playing a regulatory role in alginate biosynthesis and motility. It achieves this through direct interaction with the regulator AlgU and by influencing its transcription. Notably, alterations in the *mucA* gene have been linked to the mucoid phenotype observed in *P. aeruginosa* strains associated with cystic fibrosis [30,31].

In one NP3-sensitive clone of CF341_06, a Leu833Gln substitution was identified in the catalytic area at the C-terminal domain of phosphomannomutase AlgC. AlgC is involved in the biosynthesis of the alginate, lipopolysaccharide core region, and rhamnolipid [32]. While these substitutions may have unknown effects on protein function, they do not appear to impact the phage resistance phenotype but could increase the tolerance of the biofilms under phage attack.

Additional genetic alterations in type IV-pilus-related genes PilY [24], FhaI [33], and RluA [34] identified in phage-resistant and -sensitive clones after phage exposure are presented in Appendix A.

## 4. Discussion

An increased number of case reports on phage therapy for biofilm-related respiratory infections have been published, and all of them report repeated local administration of phages by nebulization [35,36]. The choice of lytic phage for treatment is usually based on the in vitro susceptibility testing of the pathogenic bacteria, either in planktonic or biofilm growth, although there is no agreement on which testing methods best reflect the in vivo situation and are mist suitable. It is generally accepted, and supported by in vitro experiments, that phages can reduce biofilm mass by disrupting the biofilm matrix and thereby destabilizing the biofilm structure [37]. Interestingly, however, our previously published studies in *P. aeruginosa* PAO1 biofilms grown under fluid conditions showed the opposite effects after repeated phage treatment that led to enhanced biofilm formation [9]. Our present study used a Calgary device-based static biofilm growth model, a commonly accepted method of biofilm susceptibility testing [18]. We believe this model better reflects biofilm behavior in a CF lung.

We demonstrated that repeated phage treatments led to a rapid loss of the anti-biofilm effect due to the swift development of resistance and diversification within *P. aeruginosa* biofilm populations. This aligns with the previously reviewed limitations of using bacteriophages for biofilm destruction, particularly the emergence of phage-resistant subpopulations within biofilms [37]. The review also highlights how biofilms present a challenging environment for phage therapy, with high-density biofilms and quorum sensing potentially inhibiting phage infection. These factors likely contribute to the observed reduction in phage efficacy over successive treatments in our study.

The diversification of the biofilm population under phage exposure has been described previously [38], but less is known about the dynamic of resistance development after repeated treatments. We observed a decreased biomass and cell density of the biofilm after the first 24 h of treatment compared to untreated controls but a reduced effect after two exposures and no effect of the phage treatment after the third exposure. This was observed in all the tested combinations of *P. aeruginosa* strains and lytic phages, suggesting a general response of *P. aeruginosa* biofilms to repeated phage treatments. The enhanced biomass that was previously observed in flow-cell biofilms after repeated exposure to phages was not observed in the evolution in the Calgary device. This might be due to the different growth conditions in the two biofilm models: continuous administration of nutrients in the flow-cell biofilms, while the biofilms in the Calgary device represent a static model.

Interestingly, an investigation of the sensitivity to phages of isolates obtained from the biofilm populations after single and repeated phage exposures revealed that several clones retained their sensitivity to phages. This is in accordance with previous publications showing the maintenance of sensitive colonies in the compartmentalized structure of biofilms after exposure to lytic phages [39]. Although the present study does not provide data on the localization of the sensitive colonies in the biofilm, we might speculate that they represent the biofilm subpopulation with low metabolic activity in biofilms and therefore poor expression of the phage receptor and, in consequence, maintenance of their sensitivity to phages in planktonic culture testing. However, several mutations were identified in these phage-sensitive colonies, suggesting that an adaptative process is taking place in the biofilm populations exposed to phage-induced stress.

The effect on the bacterial growth of the identified mutations conferring resistance or tolerance to the lytic phages seemed to be dependent on the genetic background of the *P. aeruginosa* strain, as a fitness cost was identified in clinical CF clones but not in clones from PAO1 and PA14 biofilms.

Clones of *P. aeruginosa* biofilm populations exposed to lytic phages NP3 or M32 showed a rapid occurrence of mutations in genes related to type IV pilus and lipopolysaccharide biosynthesis.

Interestingly, in the control experiments where the *P. aeruginosa* strains PAO1 and PA14 biofilms were grown for 72 h without phage treatment, these alterations were not observed. Here, the majority of the selected and sequenced clones did not establish any mutation except for a few clones with a mutation in the *siaB* gene. This finding suggests that this mutation likely originated earlier in the biofilm population. Thus, it remains plausible that the stress induced by phage exposure might have promoted mutations and the diversification of the biofilm populations.

Type IV pili were reported to be receptors for phiKMC-like phage vB_Pae-TbilisiM32 and mutations in *pil* genes were identified in isolates from PAO1 biofilm treated with M32. In the context of phage-resistant clones, predominant alterations were primarily localized in individual genes encoding structural proteins of the pilus, including PilX, PilE, and PilQ, or their regulatory element PilS, and functional components such as PilB, MorC, and CpdA. While the substitution in PilE might alter the protein structure to some extent with an unknown effect on phage–host interactions, the frameshift and nonsense mutations in PilX result in loss of function and likely contribute to the observed phage resistance. Similarly, the nonsense mutation of PilS results in a missing sensor kinase of the two-component system PilS-PilR that might result in a lack of PilA and type IV pili [40]. PilQ forms a homomultimeric porthole in the outer membrane [41], serving as the conduit through which the type IV pilus is extruded [42], and the Val642Gly substitution within the inner site of PilQ has the potential to interfere with phage recognition. PilB is a component of the intracellular motor sub-complex in the type IV pilus assembly machinery [24]. The observed frameshift mutation results in a truncated protein, leading to the loss of function, and may be associated with phage resistance. The substitution Asp388Ala might also affect PilB’s function and the phage phenotype due to the loss of the negatively charged functional residue.

These mutations in pilus genes might be a response to the exposure to filamentous prophages which are present in both PAO1 (Pf4) and PA14 (Pf5) and the CF clinical isolates CF 341_06 and CF 341_08 (Pf5). These filamentous prophages are induced during biofilm growth becoming superinfective [43] and thus may be selected for mutational resistance. Mutations in type IV pilus-related genes correlate to decreased twitching motility. There are over 40 genes, distributed throughout the genome, involved in the assembly and regulation of the type IV pilus system in *P. aeruginosa,* and it has been shown that type IV pili play roles in surface attachment/adhesion, cell–cell aggregation, biofilm formation, and motility. Type IV pili are important for virulence, as mutants lacking type IV pili are impaired for host cell colonization and thus less infectious [24].

How the substitution in the cAMP-phosphodiesterase CpdA contributes to the phage resistance phenotype is so far unclear, particularly as we cannot predict the effect of the substitution.

The core lipopolysaccharide was proposed to be the bacterial receptor of NP3 [11], and as expected, mutations in the *galU* gene were present in three NP3-resistant isolates. Two of those resulted in different frameshifts, likely resulting in a loss of function. In the other clone, a neutral Gly170Ala substitution in a loop close to an α-helix is unlikely to impact protein function, but this cannot be ruled out. Mutations in *galU* have been also reported in *Klebsiella pneumoniae* resistant to other lytic phages [44] as well as in phage-resistant clinical *P. aeruginosa* isolates [22]. Mutations in *galU* have been previously reported to impair the virulence of *P. aeruginosa*, confirming that development of the phage resistance occurs at a fitness cost. However, the importance of decreased virulence in the context of persistence and maintenance of the biofilm is not clear. Knockout of these genes result in a truncated core polysaccharide due to the absence of D-Glc (I–III). Priebe and Co [45] studied the role of the O standard antigen (OSA) deficiency in a murine acute lethal pneumonia model and found that *P. aeruginosa galU* mutants are attenuated in virulence but still capable of causing severe and lethal pneumonia. An increased serum and opsonization susceptibility of *P. aeruginosa* strains with OSA deficiency was reported [45].

Additionally, partial involvement was observed in other genes related to lipopolysaccharide synthesis and maintenance. This is in accordance with previously published data on the known host receptors on the bacterial surface [46]. The frameshift identified the inner membrane-located serotype-specific B-band O-antigen polymerase encoded by *wzy* [26,27] likely resulted in the loss of functionality of this enzyme, leading to alterations in the O-antigen structure. Different mutations were found in the RfaB-like glycosyltransferase that plays a role in LPS biosynthesis. Mutations in *rfaB* have been previously described in *P. aeruginosa* phage-resistant isolates, in which LPS analysis showed the absence of the core and long-chain O-antigen [28]. At least, the frameshifts and nonsense mutations in the *rfaB* gene identified in our study likely reduced or abolished the function of the protein, resulting in altered LPS synthesis that potentially may impact phage recognition. The DnpA belongs to the LmbE superfamily of hydrolases recognizing substrates containing a N-acetylglucosamine core. The LmbE family gene is less studied in *P. aeruginosa*, but it is known to deacetylate various peptidoglycan substrates [47]. Therefore, the putative loss of function by a frameshift might alter the peptidoglycan composition. However, since the loss of function of all three proteins (Wzy, RfaB and DnpA) was also likely in sensitive clones, the specific impact of these proteins on the phage susceptibility phenotype remains unclear and needs further investigation.

Mutations causing tolerance are, for example, mutations in genes involved in alginate biosynthesis such as *mucA* leading to overproduction of the polysaccharide alginate. The presence of mutations in *mucA* found both in clones from PA14 biofilms exposed to NP3 and PAO1 biofilms exposed to M32 correlated to the occurrence of the mucoid colony phenotype. We have previously reported the selection of mucoid colonies in flow-cell biofilms exposed to lytic phages [9]. The occurrence of mucoid variants after exposure to lytic phages is an old observation reported during the phage typing of *P. aeruginosa* isolates [48]. Although some mucoid clones were identified as phage-sensitive, the biofilm formed by alginate-producing strains has an increased matrix, and therefore, this can be considered as a mutation leading to the tolerance of biofilms to phage attack.

Although we did not observe enhanced biomass by crystal violet staining of the phage-treated biofilms, the mutations in *morA*, *mucA,* and *algC* would support enhanced biofilm formation. Moreover, increased intracellular levels of c-di-GMP are expected to occur in *morA* mutants, and this will lock the bacterial population in the biofilm mode of growth [49].

Phage-resistant brown colonies due to the overproduction of pyomelanin have been described previously after phage treatment [50], and the pigments have been shown to protect against oxidative stress in biofilms [51,52]. Recently, it has been shown that phage-resistant pyomelanogenic *P. aeruginosa* mutants were hypersusceptible to cationic peptides LL-37 and colistin and were more easily cleared in human whole blood, serum, and a murine infection model [53].

In two PAO1 and two PA14 clones with NP3 phage resistance and one NP3-sensitive PA14 clone, no mutations could be identified. This suggests that in these clones the resistant phenotype was most likely related to altered gene expression due to stochastic gene expression effects, with possible involvement of quorum sensing [54,55]. Stochastic gene expression in bacteria refers to random fluctuations in the expression levels of genes among individual cells within a population to enhance adaptation to environmental changes. This variability arises from inherent probabilistic processes in gene regulation and finally can result in phenotypic heterogeneity of a population manifesting in different phenotypes [56]. This mechanism is, however, difficult to elucidate experimentally, as even applying transcriptomic analysis, the stochastic effects must be determined directly and in individual cells in the population (as a snapshot of gene expression in subsets of the population). A subsequent transcriptomic analysis of a selected clone after a selection on an agar plate will therefore not reveal which genes were expressed differently to result in the visible phenotype.

While our study demonstrates the limitations in the effects of phage-based treatment of *P. aeruginosa* biofilms using multiple exposures of single phages, it is important to emphasize that the application of phage cocktails may have reduced the level of phage resistance and thus enhanced the effects of phage treatment. Further, combining the phage treatment with phage–antibiotic combinations may have had synergistic effects on the biofilm control, as previously shown [9]. However, mutations that we observed in our in vitro set-up were also reported in resistant *P. aeruginosa* isolates collected in vivo, emphasizing the relevance of our findings [57].

## 5. Conclusions

In conclusion, our study highlights the significant role of genetic alterations in the LPS and type IV pilus biosynthesis pathways, which were exclusively found in phage-resistant clones and are strongly linked to the resistance phenotype. These findings align with previous research identifying these proteins as critical factors in phage–host interactions. Additionally, our study shows a rapid decrease in the efficacy of repeated phage therapy on *P. aeruginosa* biofilms due to the development of resistance and diversification of the *P. aeruginosa* biofilm populations as shown by the accumulation of mutations in several genes and selection of alginate and pyomelanin producers. Gene alterations observed in both resistant and sensitive clones suggest an early response to phage-induced stress, potentially playing a crucial role in the biofilm’s adaptation to phage exposure. Mutations predominantly found in sensitive clones may contribute to the phenotypic diversification of biofilm populations under phage attack. This underlines the complexity of the anti-biofilm mechanisms of phages and vice versa. The influence of these adaptive mutations on population dynamics and further adaptation to phages warrants deeper investigation, particularly given the growing interest in phage therapy as a treatment strategy. Understanding these mechanisms is essential for developing effective phage-based interventions against biofilms.

## Figures and Tables

**Figure 1 microorganisms-12-01880-f001:**
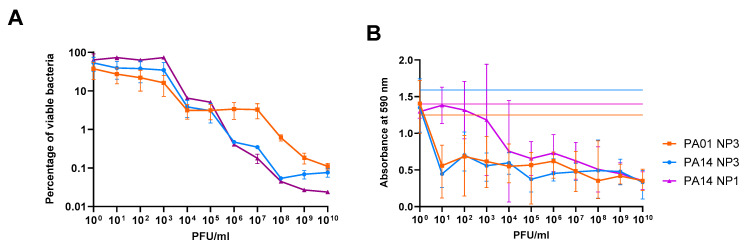
Reduction in the biofilms of *P. aeruginosa* PAO1 and PA14 depends on NP3 and NP1 phage concentrations. (**A**) The viable cells were determined as CFU/mL in four independent experiments, and the reduction was calculated as a percentage of the viable bacteria after treatment in relation to the untreated controls; means and standard error of means (SEM) are shown. (**B**) The reduction in biomass was determined via crystal violet staining of the phage-treated biofilms and untreated biofilms (indicated here as the horizontal lines). The measurements were performed 8 times, and means and standard deviations (SD) are shown.

**Figure 2 microorganisms-12-01880-f002:**
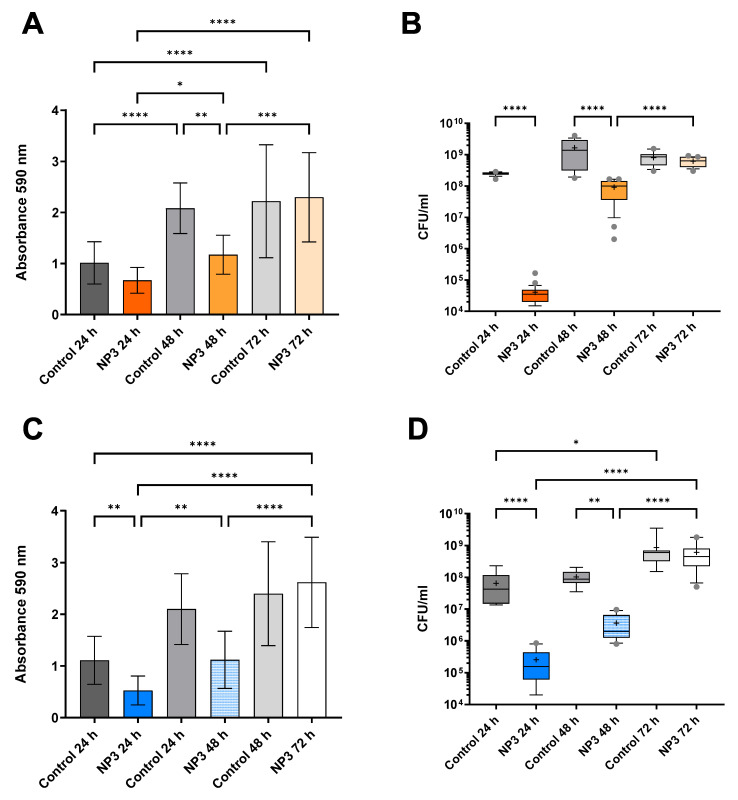
Effect of NP3 phage on PAO1 and PA14 biofilms after repeated treatments compared to the corresponding control biofilms. (**A**) PAO1 biomass is measured as CV absorbance. (**B**) Viable bacteria fraction of the resolved PAO1 biofilms as CFU/mL. (**C**) PA14 biomass measured as CV absorbance. (**D**) Viable bacteria fraction of the resolved PA14 biofilms as CFU/mL. In (**A**,**C**) mean and standard deviation (SD) and in (**B**,**D**) box blots with the 5–95%c percentile (whiskers), the 25–75% quadrille (box) with the median (line within the box) and mean (+ within the box) and outliers (grey dots) of four biological and four technical replicates are presented. Significance was assumed for *p*-values below or equal to 0.05, indicated as follows: * < 0.05, ** < 0.01, *** < 0.001, **** < 0.0001.

**Figure 3 microorganisms-12-01880-f003:**
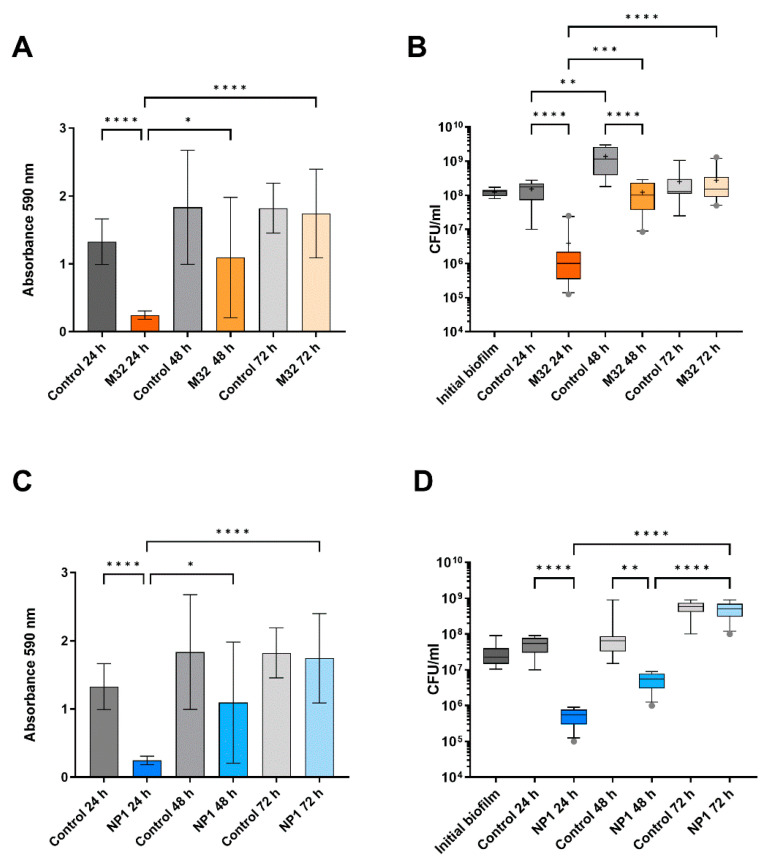
Effect of phages M32 on PAO1 and NP1 on PA14 biofilms after repeated treatments compared to the corresponding control biofilms. (**A**) PAO1 biomass is measured as CV absorbance. (**B**) Viable bacteria fraction of the resolved PAO1 biofilms as CFU/mL. (**C**) PA14 biomass measured as CV absorbance. (**D**) Viable bacteria fraction of the resolved PA14 biofilms as CFU/mL. In (**A**,**C**) mean and standard deviation (SD) and in (**B**,**D**) box blots with the 5–95%c percentile (whiskers), the 25–75% quadrille (box) with the median (line within the box) and mean (+ within the box) and outliers (grey dots) of four biological and four technical replicates are presented. Significance was assumed for *p*-values below or equal to 0.05, indicated as follows: * < 0.05, ** < 0.01, *** < 0.001, **** < 0.0001.

**Figure 4 microorganisms-12-01880-f004:**
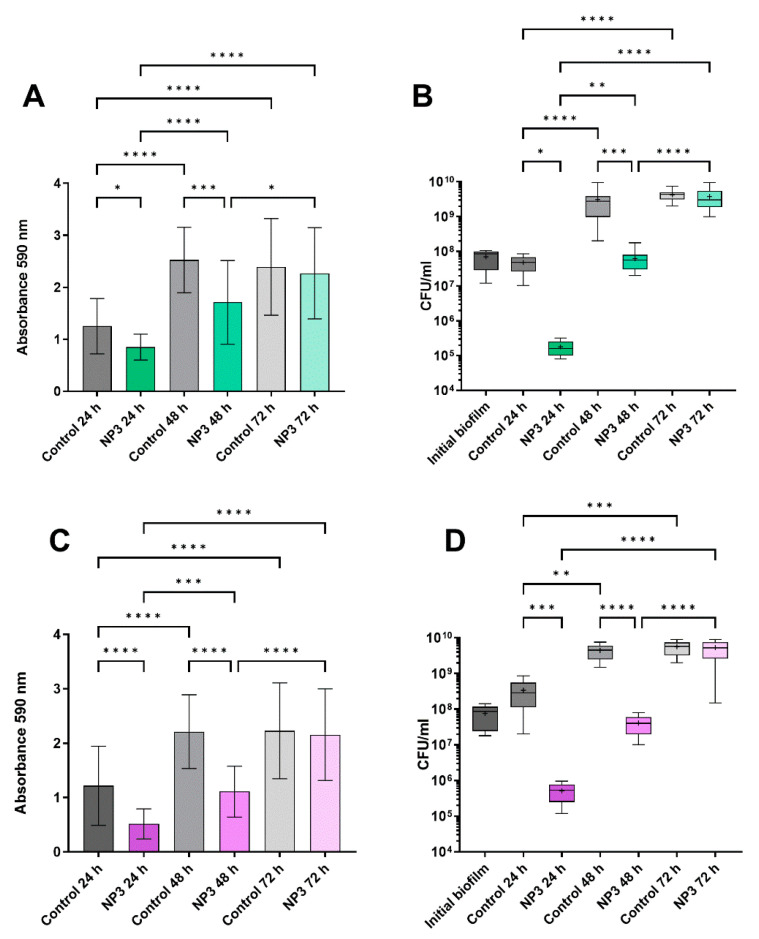
Effect of NP3 phage on biofilms of clinical isolates CF341_06 and CF341_08 after repeated treatments compared to the corresponding control biofilms. (**A**) CF341_06 biomass measured as CV absorbance. (**B**) Viable bacteria fraction of the resolved CF341_08 biofilms as CFU/mL. (**C**) CF341_08 biomass measured as CV absorbance. (**D**) Viable bacteria fraction of the resolved CF341_06 biofilms as CFU/mL. In (**A**,**C**) mean and standard deviation (SD) and in (**B**,**D**) box blots with the 5–95%c percentile (whiskers), the 25–75% quadrille (box) with the median (line within the box) and mean (+ within the box) and outliers (grey dots) of four biological and four technical replicates are presented. Significance was assumed for *p*-values below or equal to 0.05, indicated as follows: * < 0.05, ** < 0.01, *** < 0.001, **** < 0.0001.

**Figure 5 microorganisms-12-01880-f005:**
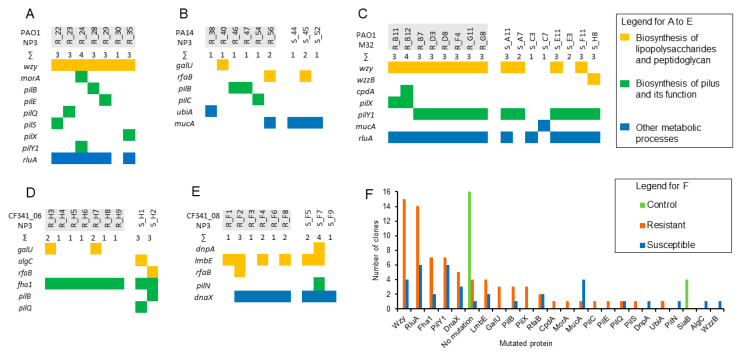
Altered genes were identified in the clones of the different strains treated with the respective phages as indicated in each diagram in (**A**–**E**). The colors in (**A**–**E**) correspond to different pathways (see legend (**F**)). The number of clones with altered genes in the controls (untreated) and treated clones exhibiting phage-resistant and -sensitive or-tolerant phenotypes. The phage-resistant clones are marked in grey. WP_0031… is the coding side WP_003138482.1 corresponding to glycosyltransferase family 4 protein.

**Table 1 microorganisms-12-01880-t001:** Phenotypic characteristics of the phage-sensitive (S) and phage-resistant (R) clones after phage treatment.

Parameter		PAO1 + M32	PAO1 + NP3	PA14 + NP3	CF341_06 + NP3	CF341_08 + NP3
Number of clones	S	8	0	10	14	16
	R	40	48	38	34	32
	Ʃ	48	48	48	48	48
Phage susceptibility	S [%]	16.7	0	20.8	29.2	33.3
	R [%]	83.3	100	79.2	70.8	66.7
	(Ø ± SD) for S [PFU/mL]	(1.43 ± 3.78) × 10^9^	/	(7.93 ± 14.7) × 10^11^	(2.94 ± 3.32) × 10^11^	(2.77 ± 2.95) × 10^11^
Twitching motility (Ø ± SD)	*d*_S_ [mm]	13.25 ± 18.83	/	21.80 ± 15.60	17.79 ± 12.21	9.64 ± 7.13
	*d*_R_ [mm]	13.33 ± 16.99	7.89 ± 13.47	6.34 ± 10.30	13.59 ± 10.04	14.31 ± 6.59
	*p*-value (S vs. R)	0.9901	/	0.0005	0.2228	0.0388
	*d*_C_ [mm]	34 ± 25.5	41.5 ± 26.7	34.6 ± 5.7	40. 3 ± 10.1	17.9 ± 10.2
	*p*-value (C vs. S)	0.051	/	<0.014	<0.0001	<0.0240
	*p*-value (C vs. R)	0.019	0.0006	<0.0001	<0.0001	0.2814
Growth rates (Ø ± SD)	Ø *µ*_R_ [1/h]	0.26 ±0.03	0.20 ± 0.04	0.23 ± 0.03	0.14 ± 0.04	0.10 ± 0.04
	Ø *µ*_S_ [1/h]	0.23 ± 0.03	/	0.18 ± 0.02	0.16 ± 0.08	0.15 ± 0.04
	*p*-value	0.0945	/	0.0196	0.6503	0.0972

Ʃ = sum, Ø = mean, SD = standard deviation, *d* = diameter, C = untreated controls (48 h biofilm growth), µ = growth rate.

## Data Availability

Data are contained within the article and Appendix A. Sequence data can be provided upon request.

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
