# Peer review of "Diversification of Pseudomonas aeruginosa Biofilm Populations under Repeated Phage Exposures Decreases the Efficacy of the Treatment"

_microorganisms, 2024, doi:10.3390/microorganisms12091880_

Round 1

Reviewer 1 Report

Comments and Suggestions for Authors

The manuscript under consideration questions if Pseudomonas bacteriophages are promising anti-biofilm agents. The papers of this kind are very important. Since the experimentation underlying the text is robust, I believe we see a major contribution to the field. I have a number of suggestions to enhance the conclusions and provide a better fit into the existing knowledge. First of all, bacteriophages vary in terms of their ability to disrupt bacterial biofilms (Knecht et al. 2020).

(1) The manuscript should explain if the studied bacteriophages were expected to be able to disrupt bacterial lipopolysaccharides. Do bacteriophage plaques feature haloes, a hallmark of LPS-targeting enzyme activity? Are there genes coding for tailspike proteins cleaving P. aeruginosa LPS (Olszak et al. 2017)?

(2) The family Myoviridae has been abolished in the year 2022 (Turner et al. 2023). Please use current ICTV nomenclature. It can be sourced from the NCBI website, please check the field ORGANISM of the respective GenBank entries.

Minor:

In the discussion, evaluate your phages in light of the known factors having impact of anti-biofilm activity in Pseudomonas phages (Chegini et al. 2020).

Throughout the text, use automatic substitution to correct spelling of the abbreviation GenBank

Lines 64-65. «However, due to the high specificity of the bacteria-phage interactions, general predictions of the effect of repeated phage therapy on biofilms are difficult to make».

Line 490. accesnumber -> accession number

Line 565. water -> distilled water

During the study, phage identity was checked by whole genome sequencing. Please let the readers know, if there were mutations in comparison to the original sequences mentioned in the paper.

Figure 5. Please make a legend with pathway colour codes. Figure 5E. Incomplete locus tag. You can use an abbreviation and explain it in the caption.

Figure S1, mention culture medium in the caption. What we see, are probably not the colonies, but the cultures.

Include supplementary sources 1 and 3 in the main reference list.

How was the host range determined? Was it a spot test on double layer agar or bacterial lawns? Was it a plaque assay?

Table S1, check Lines 6, 11-13 of the Sheet 2

Table S2, check shifted columns

Table S3, is there a possibility to add isolation year and geographic location?

Chegini Z, Khoshbayan A, Taati Moghadam M, Farahani I, Jazireian P, Shariati A. Bacteriophage therapy against Pseudomonas aeruginosa biofilms: a review. Ann Clin Microbiol Antimicrob. 2020;19(1):45. doi: 10.1186/s12941-020-00389-5. PMID: 32998720.

Knecht LE, Veljkovic M, Fieseler L. Diversity and Function of Phage Encoded Depolymerases. Front Microbiol. 2020;10:2949. doi: 10.3389/fmicb.2019.02949. PMID: 31998258.

Olszak T, Shneider MM, Latka A, et al. The O-specific polysaccharide lyase from the phage LKA1 tailspike reduces Pseudomonas virulence. Sci Rep. 2017;7(1):16302. doi: 10.1038/s41598-017-16411-4. PMID: 29176754.

Turner D, Shkoporov AN, Lood C, et al. Abolishment of morphology-based taxa and change to binomial species names: 2022 taxonomy update of the ICTV bacterial viruses subcommittee. Arch Virol. 2023;168(2):74. doi: 10.1007/s00705-022-05694-2. PMID: 36683075.

Reviewer 2 Report

Comments and Suggestions for Authors

The aim of the study was a very interesting issue of checking if Pseudomonas aeruginosa biofilm populations under repeated phage exposures change the efficacy of the treatment. I suppose a lot of effort went into the study. I am truly impressed and it's my pleasure to review this manuscript. Novelties, a significant impact of the results as well as several findings made on the basis of the obtained results are really interesting.

My only concerns are as follows:

Although the number of distinctive colonies morphotypes was subjected to a number of methodology parts, credibility of the obtained results are limited to the quite arbitrarily chosen representatives.

Lines 280-283 - the sentence is doubled.

Figure 5 F should be provided with a better quality.

Results and Discussion sections  should be separated more clearly.

A little linguistic correction would also be useful.

However, all the points mentioned above do not decrease the overall value of the research.

Comments on the Quality of English Language

A little linguistic correction would be useful.

Round 2

Reviewer 1 Report

Comments and Suggestions for Authors

The Authors answered the questions and implemented necessary changes.

There is one minor comment left. In the study,  biomass estimates were obtained with the use of a microplate reader. To my mind, the word biomass refers to weight, rather than optical density.

Comments on the Quality of English Language

/
